# High-Temperature Resistance of Anchorage System for Carbon Fiber-Reinforced Polymer Composite Cable—A Review

**DOI:** 10.3390/polym16141960

**Published:** 2024-07-09

**Authors:** Qian Liu, Ligang Qi, Anni Wang, Xiaogang Liu, Qingrui Yue

**Affiliations:** 1School of Civil Engineering, Southwest Jiaotong University, Chengdu 610031, China; liuqian@my.swjtu.edu.cn (Q.L.); yueqr@vip.163.com (Q.Y.); 2Research Center of Shanghai Carbon Fiber Composite Application Technology in Civil Engineering, China Construction Eighth Engineering Division Co., Ltd., Shanghai 200122, China; qilg@cscec.com; 3Research Institute of Urbanization and Urban Safety, School of Civil and Resource Engineering, University of Science and Technology Beijing, Beijing 100083, China; anniwang@ustb.edu.cn

**Keywords:** CFRP cable, anchorage system, high-temperature resistance, state-of-the-art review

## Abstract

Unidirectional carbon fiber-reinforced polymer (CFRP) may exhibit significant mechanical softening in the transverse direction at an elevated temperature. While significant transverse compressive stress exists on CFRP due to the clamping force from anchorage, a CFRP cable may exhibit anchorage failure when suffering an accidental fire disaster. The high-temperature resistance of a CFRP cable anchorage is critical, and clarifying the performance deterioration and failure mechanism of a CFRP cable anchorage system at elevated temperature is fundamental for clarifying its fire resistance. This paper reviews the current research status of the high-temperature resistance of CFRP cable anchorage systems from two aspects, including the high-temperature resistance of the comprising materials and the anchorage system. The reviews on the high-temperature properties of the comprising materials are summarized from two aspects. Firstly, the mechanical performance degradation of bonding epoxy resin at elevated temperatures and the effect of a filler on its mechanical–thermal properties are analyzed. Secondly, the mechanical performances of CFRP composites at elevated temperatures are summarized, with consideration of the stress state of the CFRP cable under the constraint of an anchorage device. The reviews on the high-temperature resistance of the anchorage system also include two aspects. Firstly, the temperature field solution method for the anchorage system is summarized and discussed. Secondly, the current research status of the anchorage performance at elevated temperatures is also summarized and discussed. Based on these reviews, the research shortage of the high-temperature resistance of CFRP cable anchorage systems is summarized, and further research is recommended.

## 1. Introduction

Carbon fiber-reinforced polymer (CFRP) cables are expected to reduce the self-weight effect and bring out a potential breakthrough in the spanning capacity of structures. In addition, they also have the significant advantage of corrosion and fatigue resistance in comparison to steel cable [1]. The anchorage systems for CFRP cables can be divided into three categories according to their mechanism, including bonding anchorage, mechanical anchorage, and composite bonding–mechanical anchorage [2,3,4]. A series of studies have been conducted regarding the performance of CFRP cables, including the basic mechanical properties and serviceability [5], as well as the resilience performance under impacts and high temperatures [6,7,8,9]. Yet, the high-temperature resistance of the anchorage system for CFRP cables also needs attention, which may be more critical. In CFRP cable anchorage, the mechanical properties of the comprising materials, including a resin matrix of CFRP composites and bonding materials, are sensitive to temperature and will degrade significantly at an elevated temperature. Correspondingly, the performance of the anchorage system for CFRP cables may degrade significantly due to two reasons. Firstly, the CFRP cable inside the anchorage is generally subjected to unfavorable stress, such as transverse compression due to the anchoring constraint mechanism, but the transverse compressive performance degrades more seriously than the longitudinal tensile performance at an elevated temperature [10,11,12], which can result in serious bearing capacity degradation of the anchorage system. Secondly, an epoxy bonding material is generally used in bonding anchorage or composite bonding–mechanical anchorage, and it may also significantly soften at an elevated temperature [13,14,15], leading to serious anchorage performance degradation. Furthermore, the serious bearing capacity degradation of the anchorage at an elevated temperature may cause the complete pulling out of the cable from the anchorage, resulting in the collapse of structures.

The high-temperature resistance of a CFRP cable anchorage system mainly involves two key factors. One is the temperature-dependent mechanical properties of the CFRP and bonding materials, which determine high-temperature resistance. The other is the heat transfer mechanism among the CFRP, bonding materials, and anchorage device, which determines the temperature field in the anchorage. Both of them influence the relationship between the heating time, three-dimensional temperature field, and stress field in an anchorage system, thus determining the high-temperature resistance of an anchorage system. Some research regarding the high-temperature resistance of CFRP cable anchorage systems has been conducted. Regarding the level of temperature-dependent mechanical properties of materials in anchorage, some scholars have investigated the mechanical properties of epoxy resins at elevated temperatures [13,14,15,16], the influence of filler materials on the high-temperature resistance of epoxy resin [17], the mechanical properties of CFRP at elevated temperatures [10,11,12], and the potential failure mode of CFRP composites in anchorage [18]. Additionally, regarding the level of anchorage systems, some scholars have also investigated experimental measurement and solution methods of the temperature field in anchorage systems [19], as well as the mechanical performance and failure mechanism of anchorage systems at an elevated temperature [20].

The above-mentioned research provides positive guidance on clarifying the high-temperature resistance of CFRP cable anchorage systems, but it is insufficient for the rigorous evaluation of anchorage performance at elevated temperatures. Generally, the time-dependent fire temperature field and the heat transfer characteristics of the comprising materials in anchorage systems firstly determine the temperature field in anchorage system, thus determining the temperature-dependent mechanical properties and constitutive law of comprising materials in CFRP anchorage systems. Correspondingly, the stress field in anchorage systems can be determined according to the anchoring mechanism and temperature-dependent constitutive law of the materials. Thus, at an elevated temperature, the relationship between the heating time, the three-dimensional temperature field in anchorage systems, and the stress field in anchorage systems can be formed.

In the time–temperature field relationship, the heat transfer mechanism of the materials in anchorage is critical. While heat is transferred from the surface to the interior, the CFRP composites and bonding materials in anchorage may suffer different temperatures. Accounting for the potentially different temperature-dependent mechanical properties and the constitutive law of the CFRP composites and bonding materials, the anchorage failure mechanism may be different because of the temperature softening of bonding resin or CFRP composites.

As for the temperature softening of materials, current research has relatively clarified the properties of resin [13,14,15,16] and the longitudinal tensile properties of CFRP composites [10,11], but research on the transverse properties of CFRP composites is quite insufficient [12]. However, the temperature softening of CFRP composites in the transverse direction may be more fatal, which may cause significant anchoring capacity degradation and the pulling out failure of the CFRP cable from the mechanical or mechanical–bonding composite anchorage, where the CFRP composites are subjected to a significant transverse clamping force.

Therefore, to further investigate the current research status and shortages of the high-temperature resistance of CFRP cable anchorage systems, this paper summarizes existing work from two aspects, including the high-temperature resistance of the comprising materials and the anchorage systems. Reviews on the high-temperature properties of the comprising materials are summarized from two aspects. Firstly, the mechanical performance degradation of the bonding epoxy resin at elevated temperatures and the effect of fillers on its mechanical–thermal properties are analyzed. Secondly, the mechanical performances of CFRP composites at elevated temperatures are summarized, with consideration of the stress state of CFRP cable under the constraint of anchorage devices. Reviews on the high-temperature resistance of the anchorage systems also include two aspects. Firstly, the temperature field solution methods for the anchorage systems are summarized and discussed. Secondly, the current research status of the anchorage performances at elevated temperatures is also summarized and discussed. Based on these reviews, the current research insufficiency on the high-temperature resistance of CFRP cable anchorage systems is pointed out, and the corresponding recommendations for the subsequent research are proposed.

## 2. High-Temperature Resistance of Materials in Anchorage System

The softening of bonding materials will obviously lead to degradation of bonding or composite mechanical–bonding anchorages. Comparatively, the softening of CFRP composites, especially their transverse mechanical properties, may also result in the degradation of anchorages, and even the failure of CFPR composites. Therefore, clarifying the mechanical performance degradation law of bonding materials and CFRP composites at elevated temperatures is essential for the high-temperature resistance of anchorage systems.

### 2.1. Bonding Materials and Fillers

Epoxy resin is generally used as the bonding material for CFRP cable anchorages in engineering applications, and fillers are also generally utilized to improve the bonding strength and mechanical performances of the epoxy resin [13,14,17]. Since the epoxy bonding material serves as the interfacial binder between the CFRP composites and the anchorage device, its softening at elevated temperatures will obviously influence the force transfer from the CFRP composite cable to the anchorage device, thus decreasing the anchoring capacity. Therefore, clarifying the temperature-dependent mechanical performances of the bonding material and the influence of fillers is essential for the high-temperature resistance of the anchorage systems.

#### 2.1.1. Mechanical Properties of Epoxy Resin at Elevated Temperatures

Generally, the mechanical properties of the epoxy resin will exhibit two significant degradations with elevated temperatures [21]. The first degradation occurs at the glass transition temperature of the resin *T*_g_, causing fracture of the weak chemical bonds between the molecular chains and resulting in the transition of epoxy resin from a glassy state to a rubbery state. The second degradation occurs at the decomposition temperature of the resin *T*_d_, causing fracture of the primary chemical bonds between the molecular chains and resulting in a dramatic increase in the mobility of the molecular chains, resulting in a near-complete loss of the mechanical performance of the resin. Generally, epoxy resin having higher *T*_g_ will achieve better mechanical performance at elevated temperatures [22].

At elevated temperatures around *T*_g_, the strength and modulus of the epoxy resin will significantly degrade [23,24], exhibiting obvious viscoelasticity. Additionally, the degradation of mechanical performance will accelerate after elevated temperature exceeding *T*_g_, and the stress–strain relationship of epoxy resin will exhibit significant nonlinearity [13]. Ke et al. [14] investigated the tensile performance of epoxy resin at elevated temperatures and summarized its evolution into four patterns, as shown in Figure 1. In Pattern-1, representing room temperature, the stress–strain relationship remains almost linear until fracture. In Pattern-2, representing elevated temperature lower than *T*_g_, the stress–strain relationship exhibits nonlinearity with non-negligible strength degradation. In Pattern-3, representing temperature around *T*_g_, the stress–strain relationship may exhibit an obvious platform with significant strength degradation. In Pattern-4, representing temperature exceeding *T*_g_, the degradation of strength and modulus is serious, with little retention, resulting in a near-complete loss of the mechanical performance of the epoxy resin. It should be noted that the transition process from Pattern-2 to Pattern-4 may not be obvious, which is to say that Pattern-3 may not occur clearly [14,15].

#### 2.1.2. Influence of Fillers on the Performance of Bonding Epoxy Resin

In bonding anchorages, the bonding epoxy resin first provides chemical adhesive bonding to the CFRP composites, and transfers forces from the CFRP composites into the bonding resin. Then, the forces are subsequently transferred from the bonding resin into the anchorage devices. That is to say, except for providing the chemical adhesive bonding, the bonding epoxy resin also serves as a force transfer medium between the CFRP composite cables and the anchorage devices. Thus, the mechanical properties, such as strength and modulus, can influence the mechanical performance and the efficiency of CFRP cable anchorages [25]. For improving the compressive strength and modulus of the bonding materials, the fillers are generally added into the bonding epoxy resin. Particularly, by adding a variable ratio of fillers in the bonding epoxy resin, a bonding anchorage with variable stiffness can be achieved [26,27], which can relieve the stress concentration and avoid premature CFRP cable fracture at the end of the anchorage, thus improving the anchorage efficiency. Additionally, fillers may also improve the toughness of the bonding resin [26,27,28,29,30,31].

Existing research has validated that the fillers will influence the mechanical properties of the epoxy bonding resin, which is summarized in Figure 2. Among different fillers, glass fiber powder [26], cement-based grout powder, and steel grit [28] can significantly improve both the compressive strength (*f*_c_) and the modulus (*E*_c_), quartz sand can significantly improve the compressive modulus but not obviously decrease the compressive strength [27], Al_2_O_3_ micro-balloon can improve both the tensile strength (*f*_t_) and the modulus (*E*_t_) [29], and nano-SiO_2_ powder can improve the tensile strength and fracture elongation but decreases the tensile modulus [30]. Zhou et al. [26] added glass fiber powder into the epoxy resin with a content of 40% and achieved a compressive strength improvement of about 50%. Feng et al. [27] added quartz sand into E51 epoxy resin with different contents and found that the compressive strength remained almost unchanged with increasing filler contents, but the compressive modulus increased linearly with increasing filler contents, where 80% filler content achieved a modulus improvement from 2.1 GPa to 6.7 GPa. Wu et al. [28] comparatively investigated the influence of different fillers on the compressive strength of E51 epoxy resin, including cement-based grout powder and steel grit matrix (1 mm steel grit and pyrochlore powder), and found that adding cement-based grout powder could improve the compressive strength from 108 MPa to 173 MPa, while adding steel grit matrix could achieve a compressive strength of 268 MPa. In summary, adding glass fiber powder, cement-based grout powder, or steel grit is beneficial for improving the compressive strength and modulus, among which steel grit achieves the best improvement. Adding Al_2_O_3_ micro-balloon is beneficial for the tensile strength and modulus, and smaller filler size can achieve a better improvement [29].

The influence of fillers is also related to their contents. For example, the tensile stress–strain relationship of the epoxy resin with different contents of nano-SiO_2_ powder is shown in Figure 3 [30]. As can be seen, the tensile strength gradually increases while the tensile modulus gradually decreases with increasing filler contents within 30%, but both the tensile strength and the modulus decrease with increasing filler contents over 40%. In particular, adding 60% content also seriously decreases the tensile strength, which is due to the filler agglomerate in epoxy resin, resulting in significant internal defects. In fact, high filler content will also decrease the fluidity [26,27] of epoxy resin, and thus cause perfusion difficulty and introduce defects in the bonding resin. Therefore, the optimal filler content should be investigated to achieve optimal improvement of the mechanical properties and the convenient processing in engineering application.

In addition, adding fillers can also improve the glass transition temperature of epoxy resin. Li et al. [30] added nano-SiO_2_ powder into E51 epoxy resin with a content of 40% and achieved a *T*_g_ improvement from 52.5 °C to 66.1 °C. Wang et al. [32] added Al_2_O_3_ particles (with micro-meter size and good granular composition) into epoxy resin with a content of 67% and achieved a *T*_g_ improvement from 119.8 °C to 130.6 °C. Furthermore, Wang et al. [32] also explored the influence of particle size (5.8 μm, 2.6 μm, and 0.5 μm) and filler content, and found that *T*_g_ increased continuously with more filler content and the medium particle size of 2.6 μm achieved the best improvement, as shown by the red line in Figure 4. The mechanism of the glass transition temperature improvement by adding fillers can be attributed to two reasons. Firstly, the nano or micro filler particles would have strong interactions with the resin molecules and provide chemical or physical crosslinking [33,34], thus increasing the crosslinking density of the epoxy resin and improving the glass transition temperature. Secondly, the filler particles with good granular composition would restrict the movement of the molecular chains on the interface region [35,36], thus increasing the stiffness of the molecular chains and improving the glass transition temperature.

Furthermore, adding fillers can also improve the thermal conductivity of epoxy resin [37]. Wang et al. [32] found that Al_2_O_3_ particles could improve not only *T*_g_ but also the thermal conductivity, with similar improvement trend. The thermal conductivity could be elevated to 5.45 times when Al_2_O_3_ particles with good granular composition were added with content of 67%, as shown by the black line in Figure 4. Furthermore, other scholars also found that Al_2_O_3_ fillers with larger particle size and good granular composition could achieve higher thermal conductivity [38,39]. This phenomenon is due to the fact that relatively larger particle size is less likely to agglomerate in the resin, thus forming a three-dimensional thermal conductivity grid and facilitating the heat conductivity [32].

#### 2.1.3. High-Temperature Resistance of Epoxy Resin with Fillers

Generally, higher temperature resistance requires increasing crosslinking density of epoxy resin, which is in correspondence with improving the glass transition temperature [35,36]. Existing research has confirmed that fillers such as nano-SiO_2_ powder [30,33] and Al_2_O_3_ particles [32] can improve the glass transition temperature of epoxy resin. That is to say, the high-temperature resistance of the epoxy resin with such fillers can also be improved, which has been proven by existing experimental investigation [40,41]. For example, adding 60% volume content of fire retardant and fly ash fillers was validated to achieve significantly higher mechanical properties than pure epoxy resin at 80 °C [40], and adding boron nitride nanosheets into epoxy-based thermally conductive structural film adhesive could increase the glass transition temperature from 202 to 215 °C and achieve 40% shear strength retention at 150 °C [41].

In fact, the high-temperature resistance of the epoxy resin with fillers is critical for the anchorage efficiency retention at elevated temperatures. Unfortunately, current research regarding this issue is relatively scarce. Generally, the epoxy resin with steel grit filler is most widely used as the bonding material in bridge cable anchorages. Pan et al. [42] investigated the mechanical performance of the epoxy resin with steel grit filler after suffering 240 h elevated temperature of 150 °C, and found that its compressive strength showed no significant degradation, indicating its good thermal stability and aging properties. Moreover, they also investigated its compressive strength at different elevated temperatures, and found that the compressive strength degraded softly with increasing temperatures. At room temperature, its compressive strength was about 210 MPa, while at 100 °C, its strength retention ratio was 68.4%. However, its mechanical performance evolution with elevated temperatures was not clear, as well as the critical high-temperature resistance limit. As described in Section 2.1.1, the mechanical properties, including the compressive, tensile, and shear strength and modulus, will all degrade to some extent at elevated temperatures. While adding steel grit filler can improve its compressive strength and modulus at room temperature, the quantified influence of fillers on the evolution law of the mechanical properties of the epoxy resin at elevated temperatures is still not clear.

In addition, the thermal conductivity of the epoxy resin will also be improved after adding fillers [32,37,38,39,41]. Generally, larger thermal conductivity will cause faster heat transfer from the metal anchorage devices into the bonding epoxy resin. On one hand, the faster heat transfer inside the bonding epoxy resin will result in a relatively uniform temperature field, thus decreasing the temperature peak in local epoxy resin contacting the metal anchorage device and avoiding significant softening of local epoxy resin, which may be beneficial for the anchorage degradation at elevated temperatures. On the other hand, the faster heat transfer may also cause higher elevated temperatures inside the global bonding epoxy resin in the anchorage, thus resulting in higher risk of the softening of global epoxy resin, which may be harmful for the anchorage degradation at elevated temperatures.

In summary, the influence of fillers on the mechanical properties at elevated temperatures and the thermal conductivity of the epoxy resin may have coupling influence on the high-temperature resistance of the CFRP cable anchorages. Currently, this problem has not been quantitatively investigated, and there is not a clear conclusion whether it is beneficial or not for the high-temperature resistance of anchorages.

### 2.2. CFRP Composites for Cable

Unidirectional CFRP composites comprising carbon fiber and epoxy resin are most widely used for manufacturing cables, but their anisotropic mechanical properties in the longitudinal (fiber) direction and the transverse direction are obvious. While the longitudinal tensile properties at elevated temperatures are critical for the high-temperature resistance of the CFRP composite cable body, the transverse compressive properties are also critical for the high-temperature resistance of the CFRP anchorage. This is because significant transverse compressive stress may exist on CFRP composites in the mechanical or composite bonding–mechanical anchorages, for improving the anchorage efficiency. Therefore, clarifying the longitudinal and transverse mechanical properties of CFRP composites at elevated temperatures is essential for investigating the high-temperature resistance of cable anchorages.

#### 2.2.1. Longitudinal Tensile Properties of CFRP Composites at Elevated Temperatures

Currently, the longitudinal tensile properties of unidirectional CFRP composites at elevated temperatures are relatively clear. Since the properties of the carbon fiber are relatively insensitive to temperature [43], it is widely accepted that the longitudinal tensile properties are mainly dominated by the high-temperature resistance of the resin matrix, especially dominated by the glass transition temperature *T*_g_ when significant softening of the resin matrix occurs [44,45,46,47,48,49,50,51,52,53,54]. Generally, at temperatures lower than *T*_g_, the degradation of both the tensile strength and modulus is not significant, with modulus degradation being smaller. At temperatures around *T*_g_, the tensile strength degradation accelerates, but the tensile modulus degradation is still relatively small. At temperatures over *T*_d_ until complete decomposition of the resin matrix, the tensile strength degradation continuously develops, and the tensile modulus degradation accelerates. The accelerating tensile strength and modulus degradation at temperatures around *T*_g_ is mainly dominated by the softening of the resin matrix and the fiber–resin interface [49,50,51,52,53,54], but the degradation ratios reported by different studies are not fully consistent, which may be due to the fiber probabilistic defects based on different carbon fiber manufacturers and the CFRP composite probabilistic defects based on different manufacturing quality [55,56]. It should be noted that the tensile strength and modulus at room temperature also exhibit significant discreteness due to such defects [57,58]. In addition, the final tensile strength and modulus degradation ratios after complete decomposition of the resin matrix are also not fully consistent in the reports by different studies. Theoretically, the final strength retention ratio is dominated by probabilistic fiber defects and the CFRP composite manufacturing defects, such as fiber wrinkles, and the final modulus retention ratio is mainly dominated by manufacturing defects, such as fiber wrinkles [55,56].

Anyway, such inconsistencies do not influence research on the high-temperature resistance of CFRP cable anchorages. This is because the degradation of the tensile strength and modulus of CFRP composites at elevated temperatures within *T*_g_ is generally within 20% and 10%, respectively [48,51,53], but the degradation of the transverse properties is much more significant [11,12], especially the transverse compression [12]. Thus, the high-temperature resistance of CFRP cable anchorages is dominated by the high-temperature resistance of the transverse compression of CFRP composites, which may cause the CFRP cable to pull out from the anchorages due to significant softening of the transverse compression of CFRP composites at elevated temperatures.

#### 2.2.2. Transverse Compressive Properties of CFRP Composites at Elevated Temperatures

Existing research has confirmed that the transverse compressive strength of the unidirectional CFRP composites is just about 10% of the longitudinal tensile strength at room temperature [59]. Considering the allowable clamping force on the CFRP composites in the mechanical or composite bonding–mechanical anchorages, this issue may restrict the anchorage performance, especially at elevated temperatures when the transverse compression properties significantly degrade [12]. Similarly, existing high-temperature failure of the cold-cast socketing anchorage for steel cables has also validated that the softening of epoxy resin is fatal, leading the cable body to pull out from the anchorage device. This failure mode of the steel cable anchorage occurred in Chishi Bridge (Hunan Province, China) during its construction process, where faulty welding led to burning of the HDPE jacket and the corrosion-protective oil of the cable, starting from the end of the anchorage [60]. Thus, clarifying the temperature-dependent transverse compressive properties of the unidirectional CFRP composites is essential for investigating the high-temperature resistance of CFRP cable anchorages. Unfortunately, the relevant research is quite insufficient, even for the transverse compressive properties of CFRP composites at room temperature.

From the micro-structural level, the transverse compression failure mechanism of the unidirectional CFRP composites was generally believed to be dominated by the debonding of the fiber–matrix interface according to numerical simulation utilizing Representative Volume Element (RVE) [61,62,63], and the direction of the fracture failure surface was also related to the fiber–matrix interface bonding strength [61,63]. In addition, the micro-experimental investigation on the transverse compression of the unidirectional CFRP composites also confirmed such a failure mechanism [64]. From the macro-structural level, Han et al. [65] also found that the unidirectional CFRP composites failed by longitudinal splitting in the fiber direction under transverse compression, which was consistent with the micro-structural level investigation results. Additionally, no obvious cracking in the transverse direction of fibers could be found, indicating that the fibers contributed little to the transverse compressive performance of CFRP composites. Li et al. [63] investigated the influencing parameters by numerical simulation utilizing RVE, but the experimental validation and the consideration of size-effect from the micro level to the macro level were not sufficient. In summary, the quantified influence of the dominant parameters on the transverse compression of unidirectional CFRP composites is still not clear enough. Furthermore, the enhancing effect of the anchorage constraint on the transverse compression of the unidirectional CFRP composites has also not been investigated, while the CFRP composites may be subjected to confining compression by the anchorage devices.

As for the transverse compressive properties of the unidirectional CFRP composites at elevated temperatures, existing research is rare. Generally, the transverse compressive properties should theoretically be more sensitive to elevated temperatures than the longitudinal tensile properties, which has been validated by Xu et al. [12]. In addition, the initiation of damage and cracking development mode at elevated temperatures would not be changed in comparison to those at room temperature [12,61,62]. However, the degradation law of transverse compressive properties with elevated temperatures has still not been clearly investigated; especially lacking are testing data. In addition, the compressive stress–strain relationship and its constitutive model at elevated temperatures are also not established. Furthermore, the enhancing effect of the anchorage confining constraint on the transverse compression of the unidirectional CFRP composites at elevated temperatures has also not received attention. In summary, such fundamental research needs more attention for clarifying the high-temperature resistance of CFRP cable anchorages.

## 3. High-Temperature Resistance of CFRP Cable Anchorage System

### 3.1. Temperature Field in Anchorage System

The high-temperature resistance of the structural components can be tested by the steady-state test and the transient state test. The steady-state test can clarify the mechanical performances at a specific elevated temperature. The transient state test can obtain the temperature resistance limit under specific loads and standard fire field temperature rise curve [66], thus obtaining the critical temperature value and the fire resistance time limit, making it the most widely used engineering test. In the transient state test, the temperature field inside the CFRP cable anchorage systems is not uniformly distributed, making it not consistent with the fire field temperature. Therefore, clarifying the temperature field inside the CFRP cable anchorage systems is essential for analyzing the high-temperature resistance limit.

#### 3.1.1. Measurement of Temperature Field

Steady-state tensile tests of GFRP at elevated temperatures by Rosa et al. [67] confirmed that the temperature on the surface of the testing specimen could be much lower than the environmental temperature in the heating furnace due to the hysteresis effect of heat conduction, and gradually trended to be consistent with an increase in the heating time and stabilization of the environmental temperature in the heating furnace, as shown in Figure 5. Therefore, for the transient state test of the high-temperature resistance limit of the CFRP cable anchorage systems, especially large-size anchorages for engineering cables, the temperature field inside the anchorage systems should also not be uniform, exhibiting a temperature gradient from the surface to the interior. Thus, obtaining the temperature field inside the CFRP cable anchorages is essential for determining the temperature-dependent mechanical properties of the comprising materials and giving more rational evaluation of the high-temperature resistance of CFRP cable anchorages. 

Thermocouples are mostly used in experimental measurement of the temperature field, but they are limited for obtaining the temperature field inside large-size components due to the constraints of thermocouple arrangement density and location fixing. Using thermocouple measurement, Wang et al. [68] confirmed significant temperature difference between the surface and interior center of a steel cable anchorage. Further, Zhu et al. [66] obtained the temperature field of a steel cable anchorage in the fire field by installing multiple thermocouples, as shown in Figure 6, in which the red points represent the thermocouples fixed on the steel wires inside the anchorage. The results indicated significantly different temperatures at different locations inside the anchorage. In addition, it should be noted that only part of these thermocouples successfully gave effective temperature measurement data, and some thermocouples failed during the specimen preparation and testing process, which is to say that thermocouple measurement was somewhat technically limited in practice. Although thermocouples can present actual measurements of the temperature at specific points, more thermocouples inside the CFRP cable anchorage are needed to obtain precise the temperature field in a CFRP cable anchorage. However, it is technically difficult to realize in practice, due to restrictions such as arrangement spacing and location fixing of the thermocouples. Therefore, the combination of the thermocouple measurement and the temperature field calculation is a more applicable approach, where the thermocouple measurement can be used as validation for the temperature field calculation.

#### 3.1.2. Calculation of Temperature Field

Considering the limitation of the thermocouple measurement, the calculation of temperature field by FEA is also utilized, where the thermocouple measurement by testing is used as the constraint conditions in the FEA calculation process. Kodur et al. [69,70] developed a macroscopic finite element model using the FORTRAN programming language to explore the thermo-mechanical response of FRP strengthened concrete specimens under the combined effects of fire exposure and flexural loading, and the structural analysis was carried out following the thermal analysis to evaluate the structural failure and fire resistance limits. Commercial finite element software, such as ABAQUS, can also be utilized for the three-dimensional temperature field solution of CFRP cable anchorages. For example, Zhou [71] and Liu [72] analyzed the temperature field of CFRP bar and CFRP grid-reinforced concrete beams using finite element software ABAQUS, utilizing the experimental thermocouple measurement as validation of the FEA model. During the FEA process [72], the thermophysical coefficients of CFRP, concrete, and fireproof coating were first calculated using empirical equations, including the temperature-dependent density, the thermal conductivity, and the specific heat capacity. Subsequently, according to the fire conditions in the furnace and the temperature boundary conditions of the specimen, the heat conduction analysis model was established, and the heat transfer convection coefficient and the radiation coefficient were determined. Finally, the finite element model was established and the temperature field was calculated, and the experimental thermocouple measurement was compared to validate the accuracy of the FEA temperature field calculation results. Du et al. investigated the temperature field of the steel cable anchorage with epoxy resin bonding [19] and metal bonding [73], using ABAQUS to establish the three-dimensional temperature field analysis model and reproduce the heat transfer process inside the anchorage during the temperature elevation process. The model for the steel cable anchorage with metal bonding was validated to exhibit an error within 20% in comparison to the test results [73], but the model for the steel cable anchorage with epoxy resin bonding lacked experimental validation and the temperature scope was within the glass transition temperature of the epoxy resin [19,74]. It should be noted that the three-dimensional temperature field calculation result by FEA is closely related to the thermophysical parameters of materials, which needs careful attention. Except for the empirical formula solving method, physical experiments can also be employed [19], such as measuring the specific heat capacity using differential scanning calorimetry (DSC) and thermo-gravimetric analysis (TGA), measuring the conductivity coefficient using the flash method, and measuring the heat conductivity using the heat flow method, thus giving realistic thermophysical parameters and obtaining better accuracy of the FEA model.

In summary, the FEA temperature field analysis model in combination of the test measurement validation can provide an approach for the temperature field calculation of CFRP cable anchorages. Yet, due to the restrictions of traditional thermocouple measurement, a novel temperature field measurement method for CFRP cable anchorages needs to be explored, for example, optical fiber sensing technology, which has been validated as capable to achieve stable measurement of the temperature field within 1000 °C [75,76].

### 3.2. Analysis of Anchorage Failure at Elevated Temperatures

Generally, the high-temperature resistance of cable anchorages can be investigated by transient state tests under specific loads and standard fire field temperature rise curve, and the resistance limit time can be evaluated using the criterion of allowable anchorage slip deformation [68]. Zhu et al. [66] investigated the slip evolution and failure mode of hot-cast (metal bonding) and cold-cast (epoxy resin bonding with steel grit) socket anchorages of the steel cable at elevated temperatures, and found that both of their slippage evolutions exhibited three stages, including no slippage, growing slippage, and anchorage failure, but their detailed evolution process was significantly different. In the hot-cast anchorage, the period of the non-slippage stage was the longest, and the slippage increased rapidly after its occurrence and quickly exhibited anchorage failure. In the cold-cast anchorage, the growing slippage stage was obvious, being the longest period of all stages. In addition, Ridge et al. [77] and BS EN 13411-4 [78] also confirmed such a failure mode of hot-cast and cold-cast socket anchorages.

Compared with steel cable anchorages, the high-temperature resistance of CFRP cable anchorages may be weaker. Except for the softening of the bonding epoxy resin at elevated temperatures, softening of the CFRP composites may also accelerate anchorage failure, especially when the CFRP composites are subjected to the constraint of compression forces from the anchorage devices. However, existing research on the high-temperature resistance of the CFRP cable anchorages is rare. Fortunately, some research on the high-temperature resistance of the clamping anchorages for the pre-stressed CFRP plates or tendons can be found; this is related to the failure mechanism of the CFRP cable anchorages. Jiang et al. [20] investigated the high-temperature resistance of the bonding anchorage for CFRP tendons using reactive powder concrete, and found obvious slippage failure at the anchorage end, as shown in Figure 7a. Similarly, Correia et al. [79] investigated the high-temperature resistance of the bonding–mechanical clamping composite anchorage (the glass transition temperature *T*_g_ of the bonding adhesive is about 60 °C) of the pre-stressed CFRP plates for the reinforcement of concrete structures, and found obvious slippage failure at elevated temperatures of 60 °C and 80 °C in the steady-state tests, with failure loads of 56.1% and 41.5% of that at room temperature, as shown in Figure 7b. Furthermore, in the transient state tests with temperatures elevating from 30 °C to 80 °C, in CFRP plates with pre-tension strains of 0.45% and 0.54%, both exhibited slippage failure, but the CFRP plate with pre-tension strain of 0.34% exhibited no slippage. The test results confirmed that larger clamping forces could delay slippage, but this conclusion might be limited because *T*_g_ of the CFPR plates was much higher than that of the bonding adhesive, and the elevated temperatures were much lower than the *T*_g_ of the CFPR plates.

The above research is somewhat similar to the mechanism of bonding anchorages and mechanical clamping anchorages for CFRP cables at elevated temperatures [20,79]. Theoretically, the high-temperature resistance of the CFRP cable anchorages is also determined by the thermal conductivity, the temperature field, and the temperature-dependent mechanical properties of the bonding resin and the CFRP composites. Thus, the failure of CFRP cable anchorages at elevated temperatures may include two major types according to the anchorage mechanism. Firstly, for the bonding or bonding–mechanical composite anchorages of CFRP cables, the high-temperature resistance of the bonding resin may be critical because the adhesive bonding first provides anchoring forces for the CFRP composites, and also serves as a force transfer medium from the CFRP composites to the anchorage devices. In addition, the high-temperature resistance of the CFPR composites is generally higher than that of the bonding resin, which is to say that the softening of the bonding resin will occur earlier and dominate the anchorage failure. Secondly, for mechanical anchorages or bonding–mechanical composite anchorages dominated by mechanical clamping, the high-temperature resistance of the CFRP composites may be critical. In particular, the softening of the transverse compression properties at elevated temperatures will dominate the clamping failure of CFRP composites, resulting in final anchorage failure. 

In summary, existing research on the failure mechanism of the CFRP cable anchorages at elevated temperatures is quite insufficient. To rigorously analyze the high-temperature resistance of different anchorage systems, clarifying the temperature field in the anchorage systems, as well as the temperature-dependent mechanical properties of the CFRP composites and the bonding materials, is the fundamental work that is necessary for analyzing the detailed stress distribution, as well as the mechanical degradation evolution and the failure mechanism of CFRP cable anchorages at elevated temperatures. In addition, the FEA thermo-mechanical coupling analysis can be used to analyze the stress field distribution of the anchorage systems, which has been validated as capable of analyzing the fire resistance [80] and the temperature effects [81] of the steel cables and cable-stay bridges.

## 4. Conclusions

This paper reviews the current research status of the high-temperature resistance of CFRP cable anchorages. Firstly, the high-temperature resistance of bonding epoxy resin and CFRP composites is summarized. Secondly, the temperature field calculation methods and the high-temperature resistance analysis methods for CFRP cable anchorage systems are summarized. Some conclusions can be drawn, as follows:
(1)The degradation of the mechanical properties of the boding epoxy resin at elevated temperatures is dominated by the glass transition temperature. Appropriately added filler contents, such as glass fiber powder, cement-based grout powder, and steel grit, can improve the compressive strength and modulus, while Al_2_O_3_ micro-balloon can improve the tensile strength and modulus. Furthermore, adding fillers such as Al_2_O_3_ micro-particles and nano-SiO_2_ powder can also improve the glass transition temperature, and may be beneficial for the high-temperature resistance as well. (2)The longitudinal tensile properties are relatively insensitive to elevated temperatures within the glass transition temperature of unidirectional CFRP composites. Comparatively, the transverse compressive properties are dominant over the high-temperature resistance of the CFRP cable anchorage systems. It is generally believed that the transverse compression strength of unidirectional CFRP composites at both room and elevated temperatures is dominated by the fiber–resin interfacial bonding strength, but the quantified influencing law lacks experimental validation. The degradation law of the transverse compressive properties with different elevated temperatures is also lacking, especially experimental data.(3)Traditional thermocouples are unable to measure the temperature field inside the CFRP cable anchorage systems due to technical difficulty such as the arrangement and location fixing of thermocouples. Commercial FEA software can provide an effective approach for the temperature field calculation of the CFRP cable anchorage systems, combined with careful measurement of the thermophysical parameters of the materials. Additionally, new temperature measurement technologies such as optical fiber sensing can be utilized, serving as the experimental validation of the FEA temperature field calculation.(4)Generally, the softening of the bonding resin at elevated temperatures is critical for the high-temperature resistance of the bonding and bonding–mechanical composite anchorages, while the transverse compression softening of the CFRP composites at elevated temperatures is critical for the high-temperature resistance of the mechanical and bonding–mechanical composite anchorages dominated by mechanical clamping. However, relevant experimental investigation and theoretical analysis is quite insufficient.


The recommendations for further research are presented as follows:
(1)Adding fillers may improve both the mechanical properties and the thermal conductivity of the epoxy resin, which has coupling influence on the high-temperature resistance of CFRP cable anchorage systems, which needs quantified investigation to clarify whether it is beneficial or not. Additionally, the temperature-dependent mechanical properties and the constitutive stress–strain relationship of the epoxy resin with fillers also need to be investigated.(2)Further experimental investigation is still needed to clarify the degradation of the transverse compressive properties of unidirectional CFRP composites at elevated temperatures, as well as establish a temperature-dependent constitutive model. Combination of the RVE analysis and the micro-experimental test data is also needed to clarify the quantified influence of epoxy resin’s mechanical properties and fiber–resin interfacial mechanical properties on the transverse compressive properties of unidirectional CFRP composites.(3)The failure mechanism and the high-temperature resistance of CFRP cable anchorages at elevated temperatures are quite lacking in research. Further experimental investigation and theoretical analysis is needed, where the FEA thermo-mechanical coupling analysis can be utilized.(4)The enhancing effect of the confining constraint by anchorage devices on the transverse compression of unidirectional CFRP composites at both room and elevated temperatures also needs attention, as it may be important for analyzing the failure mechanism of CFRP cable anchorages.


## Figures and Tables

**Figure 1 polymers-16-01960-f001:**
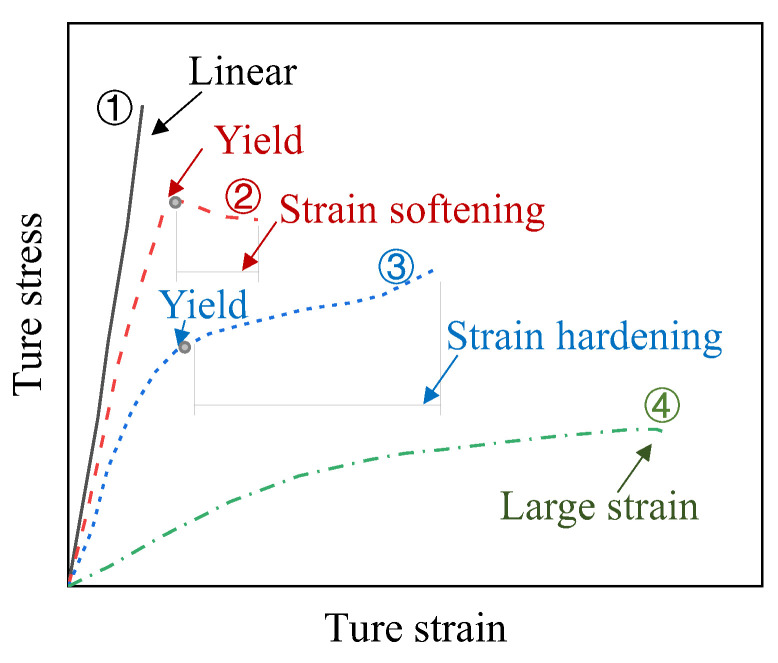
The influence of temperature on the basic mechanical properties of epoxy resin [14].

**Figure 2 polymers-16-01960-f002:**
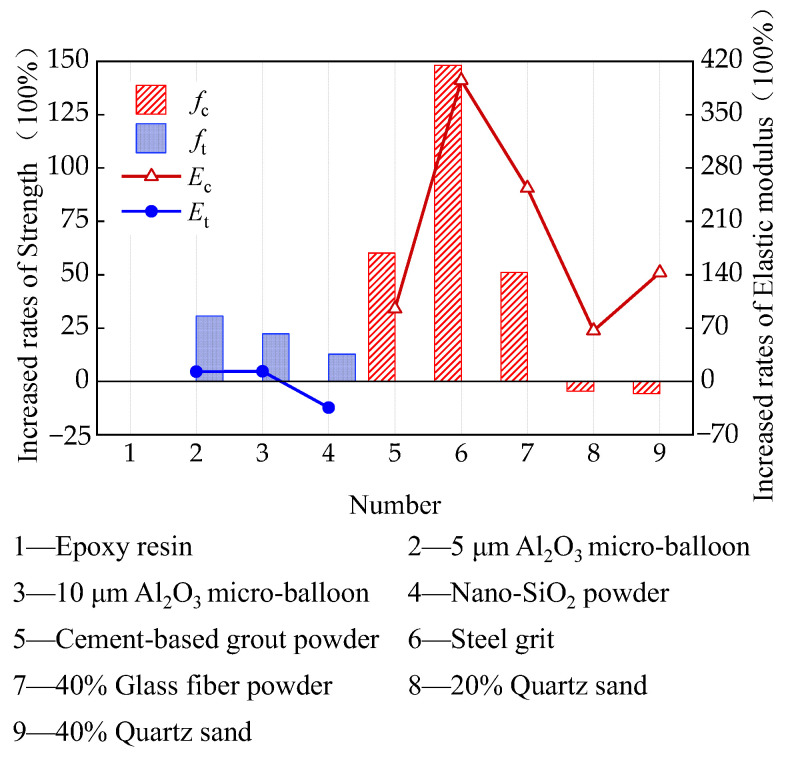
Influence of fillers on the strength and modulus of the epoxy resin [26].

**Figure 3 polymers-16-01960-f003:**
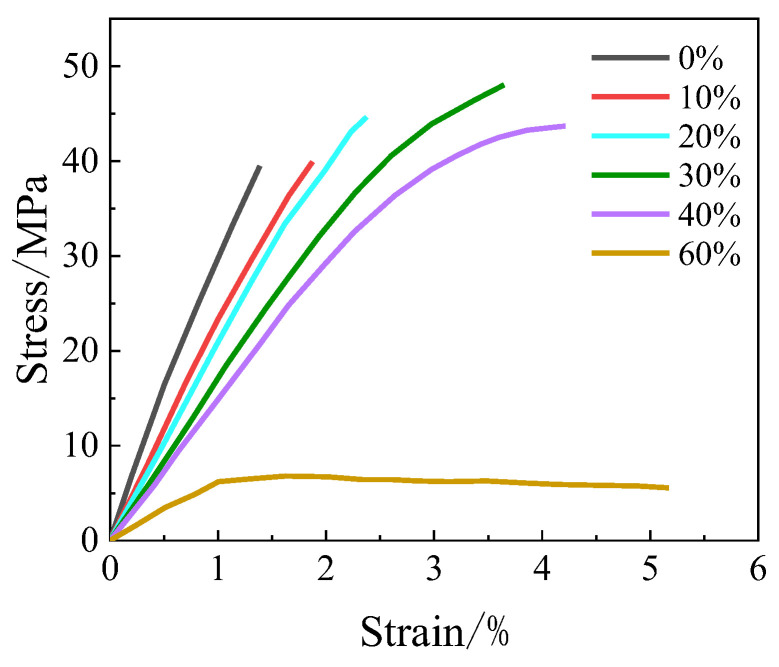
The contents of SiO_2_ vs. the tensile stress–strain relationship of epoxy resin [30].

**Figure 4 polymers-16-01960-f004:**
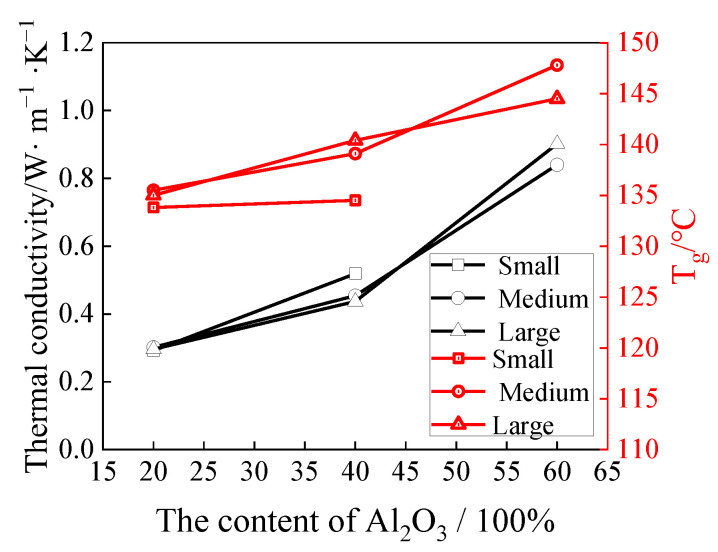
Influence of Al_2_O_3_ with different particle sizes and contents on the epoxy resin [32].

**Figure 5 polymers-16-01960-f005:**
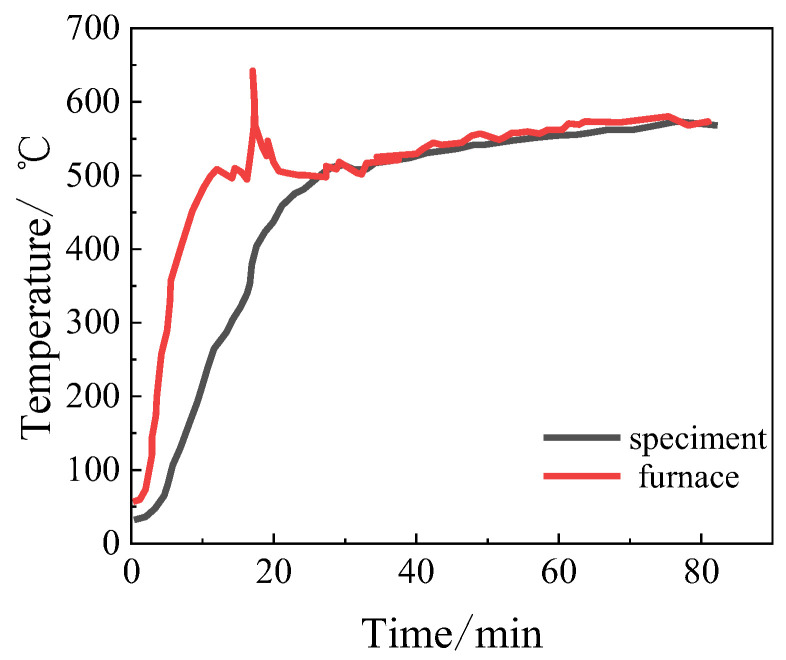
Temperature on the surface of GFRP specimen and the environmental temperature in the heating furnace [67].

**Figure 6 polymers-16-01960-f006:**
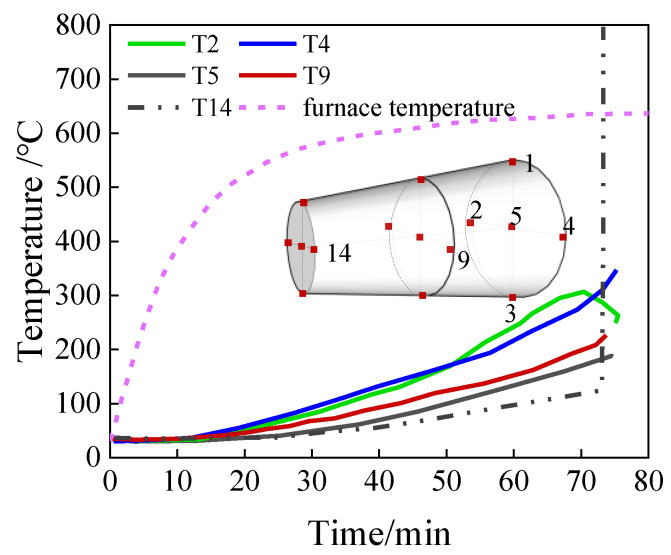
Temperature comparison of different measurement points inside the anchorage [66].

**Figure 7 polymers-16-01960-f007:**
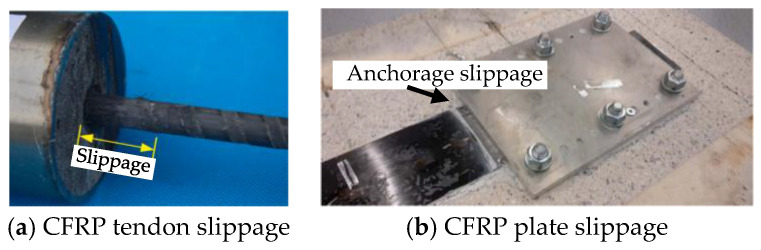
The CFRP tendon and plate anchorage slippage failure at elevated temperatures [20,79].

## Data Availability

Not applicable.

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
