# Peer review of "High-Temperature Resistance of Anchorage System for Carbon Fiber-Reinforced Polymer Composite Cable—A Review"

_polymers, 2024, doi:10.3390/polym16141960_

Round 1

Reviewer 1 Report

Comments and Suggestions for Authors

The manuscript entitled "High-Temperature Resistance of Anchorage System for Carbon Fiber Reinforced Polymer Composite Cable” investigate current research status and shortages of high-temperature resistance of CFRP cable anchorage system, this paper summarizes existing work from two aspects. Firstly, the high-temperature resistance of comprising materials in anchorage is summarized and analyzed, including bonding materials and influence of its filler, as well as CFRP composites. Secondly, the temperature field in anchorage system, as well as anchorage performance degradation and failure mechanism at elevated temperature are also summarized and analyzed. Finally, current research insufficiency on high-temperature resistance of CFRP cable anchorage systems is pointed out and recommendations for the subsequent research are proposed.

After reviewing this manuscript, I found it an interesting work. However, there are some remarks have to be considered before recommending it for publication. 

1- Some complex sentences arise through the manuscript. Shorter and more comprehensive ones will be better.

2- The conclusion section has to be rewritten.

3- Recent and relevant references published in 2023 and 2024 are recommended to be cited here.

Comments on the Quality of English Language

The manuscript entitled "High-Temperature Resistance of Anchorage System for Carbon Fiber Reinforced Polymer Composite Cable” investigate current research status and shortages of high-temperature resistance of CFRP cable anchorage system, this paper summarizes existing work from two aspects. Firstly, the high-temperature resistance of comprising materials in anchorage is summarized and analyzed, including bonding materials and influence of its filler, as well as CFRP composites. Secondly, the temperature field in anchorage system, as well as anchorage performance degradation and failure mechanism at elevated temperature are also summarized and analyzed. Finally, current research insufficiency on high-temperature resistance of CFRP cable anchorage systems is pointed out and recommendations for the subsequent research are proposed.

After reviewing this manuscript, I found it an interesting work. However, there are some remarks have to be considered before recommending it for publication. 

1- Some complex sentences arise through the manuscript. Shorter and more comprehensive ones will be better.

2- The conclusion section has to be rewritten.

3- Recent and relevant references published in 2023 and 2024 are recommended to be cited here.

Reviewer 2 Report

Comments and Suggestions for Authors

hard to compare if the article is in Chinese, and the subjects/ problem is similar
Wang A, Liu X, Yue Q. Research progress of carbon fiber reinforced polymer composite cable: anchorage system and service 516
performance . Journal of Building Structures, 2022, 43(9): 45-54. (in Chinese)

for sure some literature positions are missing
e.g. if you put in google search the title of the article "High-Temperature Resistance of Anchorage System for Carbon Fiber Reinforced Polymer Composite Cable" you can easyli find articles which were not cited by authors

due to this I do not think that the problem is oryginal or new

Also I have an impression that the work is only focusing on extracting relevant previous work without analyzing them.In my opinion authors should work more on the review - maybe understan better what does the review mean.

All figures in the article are from previous work, do the authors have a permission?

Reviewer 3 Report

Comments and Suggestions for Authors

The current paper addresses a comprehensive review of “High-Temperature Resistance of Anchorage System for Carbon Fiber Reinforced Polymer Composite Cable”. The paper is nicely written but following details needs to be addressed.

 Abstract can be written with more clarity

 Language needs to be improved, as many places long sentences are used. Also clarity in explanation is missing in few places. Please modify the content

 Long sentences. Needs to make it simple. Eg line no 68 to 74

 Scope of the review can be more elaborately given at the end of last paragraph of introduction

 Add reference for line no 107-108, 434-448, -

Sentences not clear 142-143

2.1.3. High-Temperature Resistance of Epoxy Resin with Fillers – more references needs to be added

A section on future scope can be added after the conclusion

Comments on the Quality of English Language

Language needs to be improved, as many places long sentences are used. Also clarity in explanation is missing in few places. Please modify the content

 Long sentences. Needs to make it simple. Eg line no 68 to 74

Reviewer 4 Report

Comments and Suggestions for Authors

1- The introduction provides a broad overview of the subject matter and cites relevant studies. However, the transition between discussing the general importance of CFRP cables and the specific issues related to high-temperature performance could be smoother. Including a brief explanation of why high-temperature resistance is particularly challenging for CFRP compared to other materials would enhance the context.

2- The discussion on the influence of fillers on epoxy resin properties is thorough. However, the section could be enhanced by including a comparative analysis of different filler types and their effectiveness. Additionally, mentioning potential drawbacks or limitations of using high filler content (e.g., potential processing difficulties or cost implications) would provide a more balanced view.

3- The distinction between the longitudinal and transverse mechanical properties of CFRP is well-made, but the paper could benefit from a more detailed explanation of why transverse properties are more critical in the context of anchorage systems. Providing specific examples or case studies where transverse property degradation led to anchorage failure would illustrate the practical implications more clearly.

4- 3. High-Temperature Resistance of CFRP Cable Anchorage System

4-1- The discussion of experimental measurements and finite element analysis (FEA) needs a clearer delineation of their respective advantages and limitations. A comparative analysis of these methods would provide a more balanced view and guide the reader on when each method is most appropriate.

4-2- The explanation of methods, especially regarding FEA, is somewhat superficial. A more detailed description of the specific steps, parameters used, and validation processes for FEA models would enhance the reproducibility and technical rigor of the study.

5- The conclusions section summarizes findings but should also propose specific, actionable future research directions. This could include experimental setups, new measurement technologies, or specific hypotheses to be tested, thereby providing a clear roadmap for subsequent studies.

Comments on the Quality of English Language

Minor editing of English language required

Round 2

Reviewer 2 Report

Comments and Suggestions for Authors

Accept for publication in present form